# Versatile Silver-Nanoparticle-Impregnated Membranes for Water Treatment: A Review

**DOI:** 10.3390/membranes13040432

**Published:** 2023-04-14

**Authors:** Achisa C. Mecha, Martha N. Chollom, Bakare F. Babatunde, Emmanuel K. Tetteh, Sudesh Rathilal

**Affiliations:** 1Renewable Energy, Environment, Nanomaterials, and Water Research Group, Department of Chemical and Process Engineering, Moi University, P.O. Box 3900, Eldoret 30100, Kenya; 2Environmental Pollution and Remediation Research Group, Department of Chemical Engineering, Mangosuthu University of Technology, P.O. Box 12363, Durban 4026, South Africa; 3Green Engineering Research Group, Department of Chemical Engineering, Faculty of Engineering and the Built Environment, Durban University of Technology, P.O. Box 1334, Durban 4001, South Africa

**Keywords:** disinfection, flux, fouling, membranes, silver nanoparticles, water treatment

## Abstract

Increased affordability, smaller footprint, and high permeability quality that meets stringent water quality standards have accelerated the uptake of membranes in water treatment. Moreover, low pressure, gravity-based microfiltration (MF) and ultrafiltration (UF) membranes eliminate the use of electricity and pumps. However, MF and UF processes remove contaminants by size exclusion, based on membrane pore size. This limits their application in the removal of smaller matter or even harmful microorganisms. There is a need to enhance the membrane properties to meet needs such as adequate disinfection, flux amelioration, and reduced membrane fouling. To achieve these, the incorporation of nanoparticles with unique properties in membranes has potential. Herein, we review recent developments in the impregnation of polymeric and ceramic microfiltration and ultrafiltration membranes with silver nanoparticles that are applied in water treatment. We critically evaluated the potential of these membranes in enhanced antifouling, increased permeability quality and flux compared to uncoated membranes. Despite the intensive research in this area, most studies have been performed at laboratory scale for short periods of time. There is a need for studies that assess the long-term stability of the nanoparticles and the impact on disinfection and antifouling performance. These challenges are addressed in this study and future directions.

## 1. Introduction

Water is a basic essential need for the sustenance of all life forms and the lack of safe and clean drinking water, especially for people living in developing nations, is a major concern. The United Nations Children’s Fund (UNICEF) and World Health Organization estimates that as of the year 2017, 435 million people used unimproved water sources and 144 million used surface water without treatment [1]. Water intended for human consumption should be safe, palatable, and aesthetically pleasing to prevent consumers from contracting waterborne diseases such as cholera, typhoid fever, and dysentery. These diseases are predominantly due to faecal contamination of water sources and, thus, are related to sanitation conditions. There is a dire need for access to safe and clean drinking water and for cooking and hand-washing, especially in the face of the global coronavirus pandemic.

Membrane processes are widely used in the production of clean and safe water owing to their high effectiveness, no (or less) addition of chemicals, ease of scale up, and robustness [2]. Attributes of a good membrane include a high and stable filtration flux, low filtration pressure, requires less footprint, high quality permeability, and requires minimal pre-treatment of the feed water [3,4]. Membranes are mainly classified into two types: polymeric and inorganic (metals or ceramics) membranes. Polymeric membranes are preferred due to their high flexibility and chemical stability, and are applied in pressure-driven processes such as microfiltration (MF), ultra-filtration (UF), nano-filtration (NF), and reverse osmosis (RO) [2]. Polymeric membranes are made from materials such as polyvinyl alcohol (PVA), poly acrylo nitrile (PAN), polyether sulfone (PES), and poly vinylidene fluoride (PVDF), among others. Inorganic membranes, on the other hand, are made of silica, zeolites, etc. [5,6].

The transport rate of a component through a membrane is determined by driving forces based on concentration, pressure, temperature and electrical potential gradients, and the concentration and mobility of the component in the membrane matrix [7]. The application of membranes, however, is derailed by the challenge of membrane fouling, causing loss of flux and altered rejection [4,8,9,10]. Microfiltration (MF) and ultrafiltration (UF) membrane processes are commonly used in potable water treatment and in membrane bioreactors. This is due to low energy requirement; low costs of installation and operation; effectiveness in removal of suspended matter; and appreciable removal of microbiological contaminants. However, their application for disinfection is limited by large pore sizes, hence, not an absolute barrier to microorganisms.

The main removal mechanism in MF and UF processes is size exclusion. Therefore, these processes can theoretically achieve perfect exclusion of particles regardless of operational parameters such as influent concentration, pressure, or the skills of the operator. The separation is based on the membrane pore size and the quality of product is determined by the membrane [9,10,11]. The major mechanisms of separation in MF and UF include: (a) straining, which occurs when particles are physically retained because they are larger than the pores (Figure 1a). However, this does not mean that there is 100% retention of particles larger than the membrane pore size. The interconnecting voids in membrane filters have a distribution of sizes, including some larger and others smaller than the retention rating. Therefore, particles smaller than the retention rating may be trapped in smaller passageways and larger particles may pass through the membrane in other areas [9]; (b) cake filtration, whereby particles that are small enough to pass through the membranes are retained by a cake of larger material that collects at the membrane surface during filtration (Figure 1b). This cake acts as a pre-coat filtration medium, often called a dynamic membrane since its filtering capability varies with time, growing in thickness during filtration but being partially or wholly removed by cleaning [9]; and (c) adsorption, which results when material small enough to enter pores adsorbs to the walls of the pores. If the particles and the membrane are oppositely charged, or if their zeta potentials are appropriate, the particles adhere to the membrane matrix, resulting in removal of the particles smaller than the pores of the membrane [8,11]. This means that soluble materials may be rejected even though their physical dimensions are much smaller than the membrane retention rating (Figure 1c). Adsorbed material can reduce the size of voids throughout the membrane. This, therefore, increases the ability of the membrane to retain smaller material by straining while increasing the chances of membrane fouling [9].

There is a need to enhance the membrane properties to meet needs such as adequate disinfection, flux amelioration, and reduced membrane fouling. To achieve these, the incorporation of nanoparticles with unique properties in membranes is a potential option, leading to the development of advanced ceramic and polymeric membranes with enhanced filtration performance [12]. Zwitterionic materials have also been incorporated in membranes as antifouling agents [13]. Nanoparticles of silver and copper have received considerable interest for use in water purification, especially for disinfection and for decentralized and emergency response water treatment systems. Such systems are low cost, portable, and easy to use and maintain [14,15]. Incorporation of nanoparticles into the membranes leads to increased surface per unit of mass, surface reactivity, and quantum-related effects [16]. For example, by converting bulk silver into nano-size silver, its effectiveness for controlling bacteria and viruses can be increased several times, primarily because the nano silver has an extremely large surface area, resulting in increased contact with the microorganisms [17,18]. The incorporation of nanoparticles into membranes concentrates the nanoparticles at the membrane surface where reaction occurs [15,19]. It also makes the membranes reactive instead of simply being a physical barrier, thereby performing multiple functions such as increasing water flux, improving contaminant rejection, and reducing organic and biological fouling [20,21,22].

For disinfection, metallic nanoparticles such as zinc, copper, gold, titanium, and silver have been explored. Among these, silver is the most widely studied oligodynamic material due to advantages such as its antimicrobial effectiveness on a range of microorganisms, low toxicity to human beings, and ease of incorporation into various substrates for disinfection applications. It is widely applied in domestic water filters to reduce biofouling, and in conjunction with copper ionization to prevent colonization by bacteria such as *Legionella* spp. in plumbing hospital hot-water systems [23]. Although silver nanoparticles are not toxic, especially at low concentration, their disadvantage is that accumulation in mammalian cells can lead to *argyria*, resulting from silver overload in the tissues [24,25]. Silver nanoparticles (AgNPs), when incorporated into a membrane, display strong inhibitory and biocidal properties against microorganisms that would otherwise colonize the membrane surface [26]. This is achieved using silver ions either in solution or adsorbed onto nanoparticles and the nanoparticles themselves [27]. Most studies have been conducted in the laboratory scale and for short periods of time. Long-term studies on anti-biofouling efficacy and prevention of silver loss have not been adequately covered.

Therefore, this review is critical to evaluate the potential of AgNP-coated membranes in enhancing antifouling, increasing permeability quality and flux compared to uncoated membranes. Despite the intensive research in this area, most studies have been performed at laboratory scale for short periods of time. There is a need for studies that assess the long-term stability of the nanoparticles and the impact on disinfection and antifouling performance. These challenges are addressed in this study and future directions are proposed.

## 2. Synthesis of Silver Nanoparticles

Synthesis methods of silver nanoparticles can be categorized as top–down versus bottom–up, conventional versus non-conventional, and green versus non-green (Figure 2). In the top–down approach, a large structure is gradually reduced in dimensions, until nano-size dimensions are attained after the application of severe mechanical stresses and deformations. It includes physical methods such as milling, repeated quenching, photolithography, cutting, etching, and grinding. In the bottom–up approach, nanoparticles are constructed atom-by-atom or molecule-by-molecule. Bottom–up techniques start with silver salt precursor dissolved in a solvent that is reduced in a chemical reaction and the nanoparticles are formed through nucleation and growth. They include chemical synthesis, self-assembly, and positional assembly among others [16,17,28,29].

Conventional chemical synthesis methods include the use of citrate, borohydride, organic reducers, and inverse micelles in the synthesis process. Typical reducing agents include chemical agents [14,30,31,32], plant extracts [32,33], biological agents [34], or irradiation methods [32,35,36,37,38] that provide the free electrons needed to reduce silver ions (Ag^+^) and to form AgNPs [38,39,40]. Reduction using borohydride and citrate are the most prominent. This is mainly due to the relatively high reactivity of sodium borohydride, moderate toxicity, and greater lab safety when compared to hydrogen gas and other physical methods [32,41]. Citrate is a weaker reducing agent, and the reaction requires energy that is generally applied by heating the solution. Unconventional methods include laser ablation, radio catalysis, and vacuum evaporation of metals, among others [17,38].

Green approaches use environmentally friendly agents such as sugars [32,38,42,43,44] and plant extracts such as orange peels to form and stabilize AgNPs [45,46]. However, the weakness of the green approaches is that it is more difficult to control the morphology of the produced nanosilver compared to the non-green methods [17,32,38].

## 3. Incorporation of Silver Nanoparticles in Membranes

The incorporation of silver nanoparticles into membranes for water treatment is aimed at fouling mitigation, improvement in permeability quality, and flux enhancement [47,48]. A major challenge is the dispersion of the nanoparticles in the membrane matrix. The aggregation/dispersion behaviour control is crucial [3]. Preparation of membrane composites containing silver nanoparticles can be achieved by: (i) mechanical mixing of a polymer with the nanoparticles employing mechanisms such as convection, diffusion, and shear; (ii) in situ polymerization of a monomer in the presence of the nanoparticles and in situ reduction of metal salts or complexes in a polymer [49]; and (iii) ex situ incorporation of pre-synthesized nanoparticles into the membrane [50]. Therefore, in situ synthesis requires techniques to immobilize specific functional groups on the surface of the materials, which play an important role in stabilizing and anchoring AgNPs on the filtering materials, whereas ex situ synthesis methods involve submerging or brushing membranes such as conventional ceramic filters with AgNPs solution (Figure 3). However, filters fabricated by the ex situ method sometimes lose antibacterial efficacy and clog after use [51,52].

The following methods have been employed to incorporate AgNPs in materials and are discussed in Section 3.1, Section 3.2, Section 3.3, Section 3.4 and Section 3.5.

(i) Chemical reduction of silver salts; (ii) electro-spinning; (iii) physical vapour deposition; (iv) wet-phase inversion process; (v) and dipping in colloidal silver solution or brushing with colloidal silver solution

### 3.1. Chemical Reduction of Silver Salts

This is the most-used approach to incorporate AgNPs into membrane matrices. It involves the entrapment of silver ions in the polymer chains followed by reduction with reducing agents (in situ synthesis). The advantages include: (i) the template role of the host macromolecular chains for the synthesis of nanoparticles helps improve their dispersion inside the polymeric matrix, and also partially prevents aggregation; (ii) it leads to reduced size of nanoparticles with a narrow size distribution and well-defined shape, all which are key parameters in the synthesis of nanomaterials [53]. This method has been employed to attach AgNPs on cellulose membranes [49,54], blotting paper [14], woven fabric membranes [55], polyurethane [56], and hollow-fibre microfiltration membranes [57].

To prevent NPs from aggregating, and to control the size of the final product, a stabilizing agent (capping agent) is used in the synthesis process. Agglomeration is mainly caused by excess surface energy and high thermodynamic instability of the nanoparticle surface. When solutions of silver nitrate and sodium borohydride are mixed in the absence of substances inhibiting particle growth, a fast irreversible reaction proceeds to yield a black silver metal precipitate, and the particle growth does not cease in the nanosized range. However, when the reaction is carried out in the presence of the stabilizers, the reduction process can be stopped at the stage of nanoparticle formation [15,39,58].

Reducing agents such as sodium borohydride [14,54,59], sodium citrate [56], ascorbic acid, hydrazine hydrate [60], hydroxylamine [49], and tri-octyl-phosphine [61] have been used to produce AgNPs from silver salts. The relatively high reactivity of borohydride and its non-toxicity makes the borohydride reduction the most commonly used technique to prepare AgNPs [15,62,63]. Table 1 shows the common reducing agents.

### 3.2. Electro-Spinning

Electro-spinning makes use of electrostatic forces to stretch the solution or melt as it solidifies [65]. A polymer solution or melt is placed into a syringe with a nozzle and subjected to an electric field. Under the applied electrostatic force, the polymer is ejected from the nozzle and deposited on a collector [66]. It is a simple, low-cost, and effective technology to produce polymer nano fibres. The basic setup for electro-spinning mainly used in lab scale consists of a high voltage supply, a spinneret (a syringe filled with the polymer solution or melt connected to the high voltage supply), and a grounded or an oppositely charged collector. The ejected polymer solution (or melt) becomes highly electrified by the applied high voltage (5–40 kV), which leads to the creation of an electrically charged jet that is drawn into the direction of the collector. On its way to the target, the jet is stretched and whipped, leading to the formation of nanometer-sized fibres that are collected on the target as a nonwoven fibre web. The advantages of electro-spinning are: (i) it does not affect the chemical composition of the nanoparticles or the utilized polymer; and (ii) some nanoparticles may be embedded inside the polymeric nano fibres and others attached on the nano fibres surface according to the particle size, thereby modifying the material to meet the desired outcome [67]. This technique has been used by Wang et al. [65] employing cellulose acetate solution, silver nitrate, and photo-reduction using ultra violet irradiation. Other reducing agents such as hydrazinium hydroxide and heat treatment can also be employed [68,69].

### 3.3. Physical Vapour Deposition

Physical vapor deposition (PVD) entails the use of vacuum deposition methods to deposit thin films by the condensation of a vaporized form of the desired film material onto various surfaces. The coating method involves purely physical processes such as high temperature vacuum evaporation with subsequent condensation. For the incorporation of AgNPs, the silver is heated to a point where it evaporates within the vacuum chamber and then allowed to condense on the polymer surface such as poly(vinylidenefluoride) (PVDF) and polyethersulfone (PES) [70,71]. Uniform silver deposition is achieved using electron beam bombardment of silver metal.

### 3.4. Wet-Phase Inversion Process

Phase inversion is a process whereby a polymer is transformed in a controlled manner from a liquid to a solid state through liquid–liquid de-mixing. At a certain stage during the de-mixing process, one of the liquid phases (the high polymer concentration phase) solidifies so that a solid matrix is formed [72]. Porous materials produced by precipitation from a homogeneous polymer solution are termed phase-inversion membranes. They incorporate both symmetrical (homogeneous) and asymmetrical structures. The production process consists of the following important steps: production of a homogeneous polymer solution; casting of the polymer film, followed by partial evaporation of the solvent from the polymer film; immersion of the polymer film in a precipitation solution to enable the solvent to be exchanged for the precipitation agent; and heat-setting in a bath solution in order to restructure any imperfections in the precipitated membrane film [73]. This technique has been employed to produce polysulfone UF membranes [21,74] and polyamide 6.6 membranes [75].

### 3.5. Dipping in Colloidal Silver Solution or Brushing with Colloidal Silver Solution

There are three widely used methods for impregnating ceramic pot filters with colloidal silver for disinfection: dipping the filter in a silver solution; painting the filter with silver solution using a brush; and incorporating the silver in the clay mix before firing. Ceramic filters coated with colloidal silver have been investigated for potable water treatment and disinfection applications [15,76,77,78,79,80]. The filters are mostly manufactured from locally available labor and materials such as soil, grog (previously fired clay), and water. The filter is formed using a filter press, air-dried, and fired in a flat-top kiln, at a temperature of about 900 °C over a period of 8 h. This forms the ceramic material and combusts the sawdust, flour, or rice husk in the filters, making it porous and permeable to water. After firing, the filters are cooled and impregnated with colloidal silver by painting with, or dipping in, a colloidal silver solution [81]. Recently, the application of silver nitrate to the clay, water, and sawdust mixture prior to pressing and firing the filter ceramic filter was reported, and shown to effectively reduce costs and improve silver retention in the filter [82].

## 4. Surface Characteristics Determining Membrane Fouling

Modification of membranes using AgNPs affects the morphology and structure of the membrane. Characterization is, therefore, important to verify the changes in composition, morphology, structure, and performance [83]. Membrane surface characteristics such as hydrophilicity, electrostatic charge, and roughness play an important role in membrane fouling. These characteristics often determine the interaction between the membrane and the fouling materials [84]. Membrane hydrophilicity is measured using the water contact angle measurement, electrostatic charge by zeta potential measurement, and roughness by scanning election microscope (SEM) and atomic force microscopy (AFM). To determine the amount of silver on the surface of the membrane, energy-dispersive X-ray (EDX) is employed. The main challenges in the characterization of membranes is availability of characterization equipment, especially in developing economies, and skilled personnel to operate them.

Increasing the silver film thickness results in reduced silver leaching, due to a possible enhancement of the entrapment of AgNPs in the nanocomposite matrix [70]. Increase in membrane hydrophilicity reduces susceptibility to fouling. This is because hydrogen bonds create a thin layer of bounded water on the surface of hydrophilic membrane; this layer prohibits the adhesion of hydrophobic fouling matter on the membrane surface [5,84]. Incorporation of AgNPs improves the hydrophilicity of membranes and increases of 36–77% have been reported [55,75,85]. The incorporation of AgNPs into the poly sulphone membranes introduced noticeable changes in morphology and permeate flux, especially in dense membranes [86]; however, other studies reported no noticeable morphological changes [55,87]. Table 2 shows how incorporation of AgNPs modifies these surface characteristics and, hence, affects the flux and the fouling propensity.

Figure 4 shows a postulated mechanism of antifouling using AgNPs.

## 5. Performance of Membranes Incorporating AgNPs

### 5.1. Mechanism of Antimicrobial Effect of AgNPs

The antimicrobial activity of silver has been widely exploited in the medical field because, compared with other metals, silver exhibits higher toxicity to microorganisms and lower toxicity to mammalian cells [90]. It is a powerful antibacterial agent against *E. coli*, *Staphylococcus aureus*, *Staphylococcus epidermidis*, and *Pseudomonas aeruginosa* bacteria [15,49,91,92,93,94,95]. Its use also reduces negative effects on treated water such as taste, odor, color, and formation of disinfection by products [50]. The mechanism of the bactericidal effect of silver has not been fully elucidated. The antimicrobial potential of AgNPs is influenced by the nanoparticle size, shape, surface charge, and concentration. The AgNPs display antimicrobial activity through nanoparticle attachment to microbial cells and penetration inside the cells, release of silver ions, reactive oxygen species (ROS), and free radical generation, among others [18]. These mechanisms of antimicrobial activity are summarized below and illustrated in Figure 5.

(a) Attachment and penetration by AgNPs

AgNPs, due to their size, may attach to the surface of the cell membrane, disturbing permeability and respiration functions of the cell. Smaller AgNPs have a large surface area available for interaction, which provides better contact with microorganisms [96]; hence, they have a higher bactericidal effect than the larger AgNPs [97]. The positively charged AgNPs are electrostatically attracted to the negatively charged microbial cell membrane, thus, enhancing the AgNPs attachment onto cell membranes [18].

The AgNPs can also penetrate inside the bacteria and interrupt the cellular processes [98,99,100]. According to Choi and Hu [101], smaller and uncharged AgNPs with higher surface areas interfere with the cell membrane functions by directly reacting with the cell membrane to allow silver atoms to easily enter the cells. The inhibition correlates well with AgNPs less than 5 nm, but not with the other particle sizes (10–20 nm).

(b) Release of silver ions

Antimicrobial tests on antibacterial multi-walled carbon nanotubes coated with silver nanoparticles (AgNPs) showed that Ag^+^ release was the dominant mechanism against *E. coli* and *S. aureus* due to immobilization of the AgNPs within the polymeric chains [74]. According to Mirzajani et al. [102], AgNPs interact with bacterial cell walls individually or via Ag^+^ release. Due to their nano size, they make a connection with the cell wall and generate pits on it. Thereafter, AgNPs accumulate and begin to connect more strongly with underlying layers, releasing Ag^+^ as well. AgNPs are easily dissolved and oxidized in aqueous and biological media, rendering partially oxidized NPs with chemisorbed ionic silver in such a way that both ionic and metallic silver appear to contribute to the total antibacterial activity [100,103,104]. AgNPs, due to their small sizes, have large surface areas, thereby releasing more silver ions [105,106,107,108]. The effect of AgNPs and silver ions on bacteria have been observed by the structural and morphological changes [109]. These ions interact with thiol (sulphydryl, SH) groups of the bacterial proteins, nucleic acids, and the bacterial cell membranes. When the silver ions penetrate inside the bacterial cell, the DNA molecule turns into a condensed form, due to the formation of an Ag–DNA complex and loses its replication ability, leading to cell death [92,105,110,111]. Ag^+^ ions also interact with the ribosome, thereby inhibiting the expression of the enzymes and proteins essential to ATP production [112].

(c) Action of reactive oxygen species

According to Kim et al. [113] and Kora and Arunachalam [40], the antibacterial mechanism of AgNPs is related to the formation of free radicals known as reactive oxygen species (ROS) from the surface of the AgNPs and subsequent free-radical-induced membrane damage. The uncontrolled generation of free radicals can attack membrane lipids and then lead to a breakdown of membrane function. The generation of free radicals from the surface of AgNPs inhibits bacterial growth. Inoue et al. [114] and Chang et al. [115] propose that the bactericidal effect should be considered as a synergistic action of ROS and Ag^+^ ions. Pal et al. [31] suggested that a bacterial cell in contact with AgNPs takes in silver ions, which inhibit respiratory enzymes, facilitating the generation of ROS and, consequently, damaging the cell. This disruption of membrane morphology may cause a significant increase in permeability [100,116], leading to uncontrolled transport through the plasma membrane and cell death.

### 5.2. Disinfection Performance of AgNP-Impregnated Membranes

Filtration using MF and UF membranes may be considered as a method of disinfection, but its mechanism is because bacteria are removed rather than inactivated. The separation is mostly due to size exclusion of bacteria that are larger than the pore size of the membrane. Filtration is the most-used process for the removal of particulate matter and turbidity, by water flowing through a porous media and its effectiveness in reducing microbes varies widely, depending on the type of microbe and its size. Membranes coated with AgNPs achieve complete inactivation of microbes such as *E. coli* [56,117] and *Staphylococcus aureus* [54], among others. Table 3 shows the typical performance of MF and UF membranes impregnated with AgNPs.

### 5.3. Fouling Mitigation and Flux Enhancement in AgNP-Impregnated Membranes

Membrane fouling is a major limitation in the application of membranes in water treatment. Based on fouling components, fouling can be classified into three major categories: biofouling, organic fouling, and inorganic fouling. Organic fouling is the adsorption of organic matter such as protein, grease, and humic substances onto the membrane surface. The adsorbed substances could be hydrophilic or transphilic in nature. Inorganic fouling, on the other hand, results from the deposition and accumulation of inorganic matter and other precipitates such as metallic hydroxides and silica on the surface of the membrane.

Biofouling is considered the most complicated category and seriously hampers the application of membrane processes [120]. It results from the accumulation of organics, biofilm formation, and the regrowth of microorganisms on the membrane surface [121]. However, it is usually controlled by pre-treatment of feed or chemical cleaning of the membranes during backwash. Although feed pre-treatment can be an effective form of biofouling control, many polymeric membranes cannot withstand the corrosiveness of chemical cleaners. Incorporation of antimicrobial nanomaterials into membranes, therefore, offers a potential solution to biofouling control [3].

The flux recovery ratio (FRR) of membranes is used to assess the antifouling capability, and a high FRR% indicates a better fouling resistance for the membrane. Studies have shown higher FRR for membranes containing AgNps compared to the unmodified membranes. For instance, polyamide 6,6 membranes containing Ag–graphene oxide had an FRR of 60% compared to the unmodified membranes (25%) [75]; the FRR of ceramic Ag-coated membranes was 80% and 35% for uncoated membranes [122]; and the FRR of a Ag@MOF-PVDF membrane was 95.7% [87]. Other performances reported include biofouling reduction by 80.74% using the Bradford protein assay [123], 94% reduction attachment of *E.coli* and *P. mendocina* [21], and no bacterial attachment for 9 weeks [124].

Other studies have used flux enhancement as an indicator for antifouling capability. A study reported a significant increased flux recovery in AgNP-coated membranes [125], and decreased flux decline in coated membranes [119,126,127]. The AgNPs have also been applied as surface coatings to disinfect materials, resulting in a 99.25% attachment of *E. coli* and 99.91% in *S. aureus* [128]. A study aiming to mitigate membrane biofouling under high mixed liquor suspended solid showed that the cake layer resistance coefficient of the unmodified membrane was 2.7 times higher than that of the AgNPs MF after the 60 day operation [57]. This demonstrated that the antimicrobial properties of the AgNPs resulting from the gradual release of ionic silver are effective in reducing intrapore biofouling in the membranes [57,86]. Table 4 shows typical reduction in fouling in AgNP-impregnated membranes.

### 5.4. Long-Term Performance of AgNP-Coated Membranes

Although AgNP-coated membranes demonstrate great potential in water treatment, the long-term stability of the AgNPs and the membranes is still a matter of concern. This is because most studies have been performed for short periods of time on a laboratory scale and, therefore, there is limited information regarding the long-term performance of these filters. Questions on the silver elution with time, and how this affects the disinfection and antifouling performance, continue to arise. The loss of the silver can be due to leaching of silver ions or even the release of attached nanoparticles from the membrane matrix due to continued use [131]. It is, therefore, necessary that the AgNPs be well-anchored in the polymer matrix. On the other hand, leaching of very small amounts of Ag ions can impart secondary disinfection of the treated water to prevent bacterial regrowth [132].

However, a few studies demonstrated that AgNP-coated membranes can be used for a relatively long time without significant reduction in performance. For instance, silver-embedded ceramic tablets have been used for one year with 100% *E*. *coli* reduction [117]; Park and colleagues reported no bio-fouling for membranes coated with AgNPs and polydopamine for 180 days [123]; while Zhang and colleagues reported no bio-fouling in 63 days [124]. Regarding the elution of silver ions, Mecha and colleagues observed that there was minimal elution in 90 days [132]. A similar performance was reported for ceramic membranes used for one year [117]. Table 5 shows studies on long-term performance of AgNP-coated membranes.

## 6. Future Perspectives

Despite the promising performance of the AgNP-coated membranes in water treatment, there is still need for further studies to address the challenges arising. Firstly, it is the long-term stability of the membranes. To address this, there is a growing trend of including multiple metal oxide nanoparticles such as copper, cobalt silver, titanium dioxide, and graphene oxide, as well as metal organic frameworks (MOFs); increased effectiveness has been reported as well as reduced leaching [22,25,75,87]. This also reduces the amount of silver used (which is a precious metal) without compromising the antibacterial and antifouling performance. In addition, no silver leaching was reported and this was attributed to the graphene oxide providing numerous anchor points for the AgNPs [75]. The MOFs have a large specific surface area and are used as modifiers in polymer membranes to improve mechanical strength, hydrophilicity, and antifouling capacity [87]. Novel methods of developing more reactive membranes containing AgNPs are also being explored and results show that they have great potential for long-term biocidal capability (four months) with minimal silver loss [85]. Secondly, as the use of this method for water treatment matures, there is need to move from laboratory scale studies to pilot scale and eventually large-scale application. This will facilitate the uptake of the technology from point of use (household) systems to community-scale systems to allow for ease of operation, maintenance, and management. Thirdly, development of large-scale systems requires further studies on the incorporation of the nanoparticles on large membrane surfaces to address the challenge of nanoparticle agglomeration and dispersion, membrane cleaning, and mass transfer limitations. In line with this, it is necessary to have more studies conducted for long periods of time and under real-world conditions to facilitate commercialization.

## 7. Conclusions

The use of MF and UF processes for potable water treatment has been on the rise largely due to them being less energy intensive. However, these processes are not absolute barriers to microorganisms and, hence, may compromise the quality of treated water. Improvement in the disinfection efficacy can be achieved by incorporating silver nanoparticles. The AgNPs also significantly reduce the biofouling and increases permeate flux and quality. Presently, there are minimal studies on the long-term performance of the AgNP-coated membranes. Such studies are important to determine the robustness of such systems and suitability for large-scale operation. Evaluation of silver leaching is paramount, since it may pose a health risk to consumers if high concentrations of silver are leached or result in decreased antimicrobial performance, reducing membrane lifespan. In this regard, there is need for more long-term studies targeting not only disinfection capability, but also silver elution over time and how this effects the disinfection, flux, and fouling of the membranes.

## Figures and Tables

**Figure 1 membranes-13-00432-f001:**
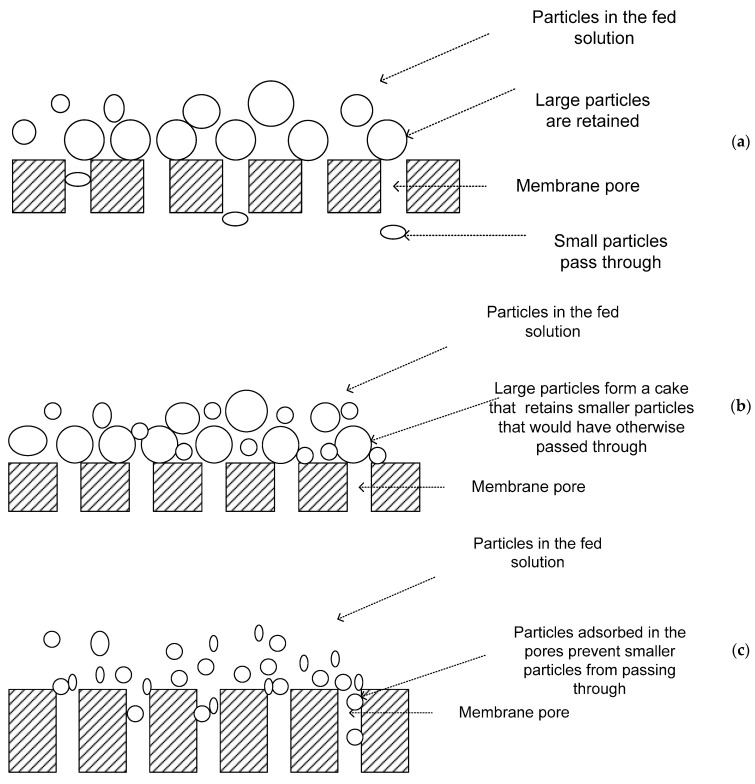
(**a**–**c**) Mechanisms of microfiltration and ultrafiltration.

**Figure 2 membranes-13-00432-f002:**
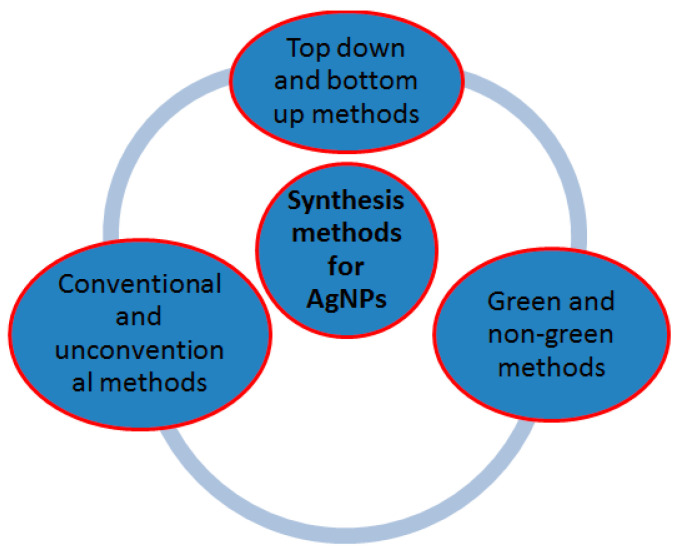
Synthesis methods for silver nanoparticles.

**Figure 3 membranes-13-00432-f003:**
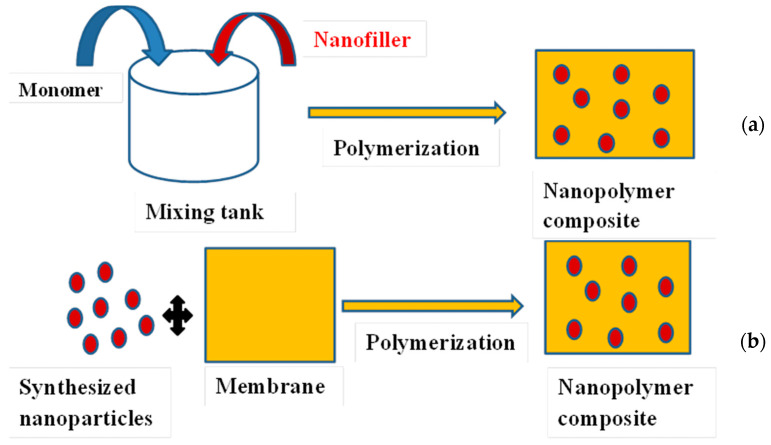
Schematic representation of in situ (**a**) and ex situ (**b**) polymerization.

**Figure 4 membranes-13-00432-f004:**
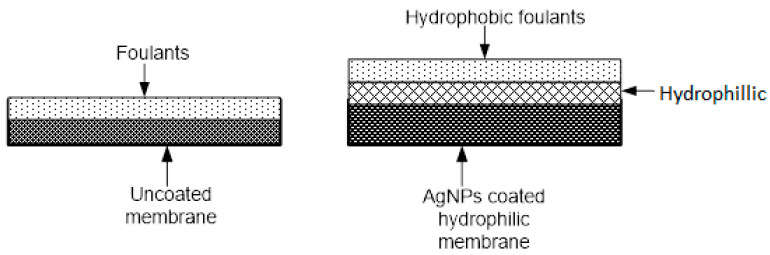
Mechanism of antifouling in uncoated and AgNP-impregnated membranes. Adapted from [5].

**Figure 5 membranes-13-00432-f005:**
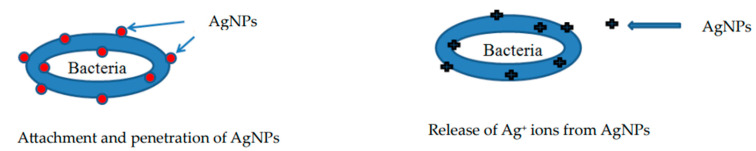
Mechanism of antibacterial action of AgNPs.

**Table 1 membranes-13-00432-t001:** Common reducing agents and the reaction conditions, adapted from Bonsak [62] and Rana et al. [64].

Reducing Agent	Temperature (°C)	Rate
Organics, alcohols, polyols	≥70	Slow
Aldehydes, sugars	<50	Moderate
Citrate	>70	Moderate
Hydrazine, H_2_SO_3_, H_3_PO_2_	Ambient	Fast
NaBH_4_, boranes, hydrated e^−^	Ambient	Very fast

**Table 2 membranes-13-00432-t002:** Characterization of membranes impregnated with AgNPs.

Property	Improvement	Reference
Hydrophilicity	Increased hydrophilicity by 36%	[85]
	Increased hydrophilicity by 46%	[75]
	Increased hydrophilicity by 77%	[55]
	Water permeability increased from 214 L/m^2^h to 1651 L/m^2^h Contact angle reduced from 80 ± 2° to zero	[88]
	The pure water flux increased from 85 L/m^2^h to 157 L/m^2^h	[87]
	The contact angle decreased from 62.8° to 54° for unmodified and Ag-ZnO modified membranes, respectively	[89]
Surface morphology	Nanoparticles uniformly distributed on the surface of the membranes and no significant difference in roughness	[55,87]
Surface charge	Membrane charge density increased 15.6-fold due to the sharp-tip morphology of the triangular silver nanoparticles forming “hot spots” on the membrane surface	[85]

**Table 3 membranes-13-00432-t003:** Performance of AgNP-impregnated membranes in inactivation of microbes.

Membrane Material	Target Microbes	Disinfection Efficiency	Reference
Ag ceramic tablet	*E. coli*	100%	[117]
Cellulose filter paper	*E. coli*	100%	[118]
Blotting paper	*E. coli* and *Enterococcus faecalis*	3–6 log removal	[14]
Cellulose membranes	*E. coli*	100% inactivation	[49]
Woven fabric membranes	*E. coli*	3 log removal	[55]
Bacterial cellulose	*Escherichia coli* and *S. aureus*	99.7% and 99.9% reduction in *E. coli* and *S. aureus*, respectively	[54]
Polyurethane foams	*E. coli* (10^5^ CFU/mL)	100% inactivation	[56]
Ceramic filters	*E. coli*	97.8–100% inactivation	[78]
Polysulfone UF membranes	Bacteriophage (5 ± 0.2 × 10^5^ PFU/mL)	5 log removal	[21]
Ceramic filter	*E. coli*	100% inactivation	[93]
Ceramic filters	*E. coli*	5.9 LRV with AgNPs; 3.05 LRV without AgNPs	[119]

LRV: log removal value.

**Table 4 membranes-13-00432-t004:** Antifouling performance of AgNP-coated membranes.

Membrane Material	Type of Fouling	Performance	Reference
Pristine membrane, SPAES/PIN-PEM	Biofouling using Bradford protein assay	80.74% biofouling reduction	[123]
Hollow-fibre MF	Biofouling under mixed liquor suspended solid (MLSS)	Cake layer resistance of the unmodified membrane 2.7 times that of the AgNPs MF in 60 days	[57]
Ceramic membranes	Bovine serum albumin	FRR increased from 35% (uncoated) to 80% (coated)	[122]
Poly ether sulfone (PES) MF membrane	Bacteria (10^7^ CFU/mL)	Flux increase (31%); FRR of 98.2% against bio fouling	[85]
Ag@MOF-PVDF membrane	Biofouling using *S. aureus*	95.7% FRR of coated membranes	[87]
Polyamide with Ag and grapheme oxide	Biofouling using *E. coli*	Flux increase (135%); 76–37% irreversible fouling reduction	[75]
Polyamide thin-film composite	2,4-dichlorophenol organic fouling	Flux recovery increased from 64% to 95%	[89]
Poly sulfone UF	Biofouling using *P. mendocina*	94% reduction attachment of *E. coli* and *P. mendocina*	[21]
Poly ether sulfone (PES)	Biofouling using *E. coli* and *P. aeruginosa*	No bacterial attachment for 9 weeks	[124]
Chitosan membrane	Biofouling using *E. coli* and *Pseudomonas* sp.	Reduced attachment for 10 days	[129]
Polypropylene membranes	Biofouling using *E. coli* and *S. aureus*	No bacterial attachment for 12 days; flux recovered in coated membranes by physical cleaning	[125]
Amicon bench-scale dead-end UF cells	Biofouling using *P. aeruginosa*	Decreased flux decline in coated membranes	[126]
Poly ether sulfone (PES)	Biofouling using *E. coli*	Flux decline was 3.7% for the coated membranes and 12.2% for the unmodified membrane	[127]
Polysulfone membranes	Biofouling using *E. coli*	Bacterial detachment ratio of 75% for coated membrane; 18% for the unmodified membrane	[130]
Ceramic filters	Biofouling using *E. coli*	Increased permeate flux	[119]
Leather	Biofouling using *E. coli* and *S. aureus*	Reduced *E. coli* attachment by 99.25% and *S. aureus* by 99.91%	[128]

FRR: flux recovery rate.

**Table 5 membranes-13-00432-t005:** Long-term performance of AgNP-coated membranes.

Membrane Material	Period	Performance	Reference
Pristine membrane, SPAES/PIN-PEM	180 days	No biofouling	[123]
Poly ether sulfone	120 days	93% biocidal activity after four months of use and 14% silver loss in 14 days	[85]
Woven fabric membranes	90 days	Minimal silver elution 0.002–0.018 mg/L	[132]
Ceramic membrane (Ag-ceramic tablet)	365 days	100% *E*. *coli* reduction; Ag leaching < 20 μg/L	[117]
Poly ether sulfone (PES)	63 days	No biofouling	[124]
Hollow fibre MF	60 days	Flux of the AgNPs MF decreased 59.7%, and in the unmodified membrane dropped 81.8%	[57]
Polyamide	28 days	2 log reduction of *E. coli* and 3 log reduction of *S. aureus*	[133]

## Data Availability

Not applicable.

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
