# Peer review of "Versatile Silver-Nanoparticle-Impregnated Membranes for Water Treatment: A Review"

_membranes, 2023, doi:10.3390/membranes13040432_

Round 1

Reviewer 1 Report

Title: Versatile silver nanoparticles impregnated membranes for water treatment: a review.

Recommendation: Minor revisions needed as noted.

1. The need for the current review, its application and usefulness needs to be explained in the end of the introduction part.

2. Must need careful revision of the manuscript, for grammatical structural, and typo mistakes.

3. What are the advantages and limitations of using silver nanoparticles in water treatment?

4. What are the key challenges associated with the synthesis and characterization of silver nanoparticles impregnated membranes?

5. How effective are silver nanoparticles impregnated membranes in removing contaminants from the water?

6. What are the future prospects and research directions for silver nanoparticles impregnated membranes in water treatment?

7. How does this review contribute to the existing literature on water treatment?

8. Check the references carefully.

Author Response

Manuscript title: "Versatile silver nanoparticles impregnated membranes for water
treatment: a review

Manuscript ID: membranes-2295201

Dear MDPI, Membranes Editorial Office,

We would like to express our gratitude for the comments that reviewers provided concerning our first submission of the manuscript and the opportunity you have given us to revise the manuscript. We have carefully read the comments and revised our manuscript accordingly. The corrections made are highlighted in yellow in the revised manuscript.

Please find a summary of the comments and the actions we took:

Reviewer 1

Comment

Action taken

Section

The need for the current review, its application and usefulness needs to be explained in the end of the introduction part.

Thank you for your comment. We have included a statement at the end of the introduction stating the need and usefulness of the current review.

Last paragraph of the introduction, line 136 to 141

Must need careful revision of the manuscript, for grammatical structural, and typo mistakes.

Thank you for your comment. The manuscript has been checked for grammatical, structural and typo mistakes.

The whole document

What are the advantages and limitations of using silver nanoparticles in water treatment?

Thank you for your comment. The advantages, and limitations of using silver nanoparticles have been stated in the introduction.

Introduction, Pg 4, line 119 to 127

What are the key challenges associated with the synthesis and characterization of silver nanoparticles impregnated membranes?

The major challenges associated with the synthesis and characterization of silver nanoparticles impregnated membranes have been mentioned in the manuscript

Page 5, Section 3.0, line 188 to 189

Page 8, Section 4.0, line 325 to 327

How effective are silver nanoparticles impregnated membranes in removing contaminants from the water?

The effectiveness of AgNPs coated membranes in water disinfection targeting various microbes and membranes has been shown in the manuscript

Page 11-12, Table 3

What are the future prospects and research directions for silver nanoparticles impregnated membranes in water treatment?

The future prospects and research directions for silver nanoparticles impregnated membranes in water treatment have been shown in the manuscript

Page 14, section 6.0, line 512-521

How does this review contribute to the existing literature on water treatment?

This review addresses major challenges in the application of membranes in water treatment and proposes ways to overcome them

Introduction, line 136 to 141

Section 6.0, line 512-521

We are grateful for the valuable comments, and we hope we have addressed them adequately for the paper to be considered for publication.

Kind regards,

Martha N Chollom

Reviewer 2 Report

Owing to the limitation of microfiltration and ultrafiltration membranes in removing smaller matters and harmful microorganisms, incorporating silver nanoparticles into the membranes enhance the performances of the membranes such as enhanced antifouling property, increased rejection performances, and improved flux. In this review, the authors have summarized recent developments towards the silver nanoparticles incorporated membranes in water treatment. Various synthesis methods of silver nanoparticles including both bottom-up approach and top-down approach are discussed. Besides, surface characteristics of a membrane in reducing membrane fouling are also included in the review. In terms of performances, the authors have discussed on the mechanism of antimicrobial effect of silver nanoparticles, disinfection performances, and also long-term performances of the membranes incorporating silver nanoparticles. The challenges and future directions in the research area are being included. This work provided an overview in synthesizing membranes with enhanced performances, disinfection, and anti-microorganisms abilities by incorporating silver nanoparticles. A review with below suggestions are listed:

 Suggestions:

1.     Synthesis of silver nanoparticles

a.      The authors mention borohydride and citrate are the most prominent used reducing agent, but borohydride is a strong reducing agent and citrate is a weaker reducing agent. The authors may justify this in the manuscript.

b.     Page 5, Figure 2. Instead of listing out the classification of the synthesis methods, the authors are suggested to branch out the details of each classification.

2.     Incorporation of silver nanoparticles in membranes

a.      The authors are suggested to include the mechanism of mechanical mixing in Figure 3 since it is mentioned in the manuscript.

b.     Page 6, line 203. The authors are suggested to list out the incorporation methods which will be included in this review instead of just refer to the title of the subsection for better understanding.

3.     Surface characteristics determining membrane fouling.

a.      Page 9, Table 2. The table is suggested to modify to ensure the column title and the contents separate properly.

4.     Performance of membranes incorporating AgNPs

a.      Page 12, line 429. Since there are three types of fouling, the authors are suggested to introduce both organic and inorganic fouling.

5.     It is advised to improve the quality of the figures in the manuscript.

6.     The authors are suggested to include more recent papers as the references. Besides, please include the following articles to broaden the perspective of this work: ACS Appl. Polym. Mater., 2021, 3, 9, 4390-4412 and Chem. Eng. Res. Des., 2021, 172, 135-158

7.     The authors are suggested to check the consistency of the format.

a.      There is double word spacing (e.g., Page 1, line 43; Page 2, line 88; etc.)

b.     The punctuations in the manuscript shall be revised properly (e.g., Page 8, line 306; Page 8, line 293; etc.)

c.      The caption shall be on the top of the table while at the bottom of a figure. (e.g., Page 9, Figure 4; etc.)

d.     The full name and the abbreviation must have indicated clearly.

8.     The citation format is not consistent. The authors are suggested to revise thoroughly on the references format for every citation.

a.      The format for the title of the journal shall be consistent, either capitalize the first alphabet of the first word only or of every word.

Author Response

Manuscript title: "Versatile silver nanoparticles impregnated membranes for water
treatment: a review

Manuscript ID: membranes-2295201

Dear MDPI, Membranes Editorial Office,

We would like to express our gratitude for the comments that reviewers provided concerning our first submission of the manuscript and the opportunity you have given us to revise the manuscript. We have carefully read the comments and revised our manuscript accordingly. The corrections made are highlighted in yellow in the revised manuscript.

Please find a summary of the comments and the actions we took:

Reviewer 2

Reviewer’s comment

Action taken

Section

The authors mention borohydride and citrate are the most prominent used reducing agent, but borohydride is a strong reducing agent and citrate is a weaker reducing agent. The authors may justify this in the manuscript.

Thank you for the comment. The main attributes of sodium borohydride and citrate have been provided in the manuscript

Page 5, section 2.0, line 166-173

Page 5, Figure 2. Instead of listing out the classification of the synthesis methods, the authors are suggested to branch out the details of each classification.

The classification of the synthesis methods is listed in section 2.0. Figure two was used for illustration purposes.

Page 4, Section 2.0, line 145 to 147

The authors are suggested to include the mechanism of mechanical mixing in Figure 3 since it is mentioned in the manuscript.

Mechanisms have been mentioned in the manuscript

Page 5, section 3.0, line 192 to 193

Page 6, line 203. The authors are suggested to list out the incorporation methods which will be included in this review instead of just refer to the title of the subsection for better understanding.

The incorporation methods have now been listed in the manuscript.

Page 6, Section 3.0 , line 210 to 214

Page 9, Table 2. The table is suggested to modify to ensure the column title and the contents separate properly.

The table have been modified to ensure column title and contents separates properly

Page 9, Section 4.0, Table 2

Page 12, line 429. Since there are three types of fouling, the authors are suggested to introduce both organic and inorganic fouling.

The three types of fouling (organic fouling, inorganic fouling and bio fouling) have been introduced in the manuscript.

Page 12, Section 5.3, line 443 to 449

It is advised to improve the quality of the figures in the manuscript.

The quality of the figures has been improved

The whole document

The authors are suggested to include more recent papers as the references. Besides, please include the following articles to broaden the perspective of this work: ACS Appl. Polym. Mater., 2021, 3, 9, 4390-4412 and Chem. Eng. Res. Des., 2021, 172, 135-158

Recent papers have been included.

Page 4, section 1.0, line 104 to 105

The authors are suggested to check the consistency of the format.

The constituency of the manuscript format has been verified.

The whole document

The caption shall be on the top of the table while at the bottom of a figure. (e.g., Page 9, Figure 4; etc.)

The captions for the tables and Figures have been edited as required.

The whole document

The full name and the abbreviation must have indicated clearly.

Thank you for your comment. We have acted accordingly.

The whole document

The citation format is not consistent. The authors are suggested to revise thoroughly on the references format for every citation.

The citations have been verified and updated accordingly.

The whole document

The format for the title of the journal shall be consistent, either capitalize the first alphabet of the first word only or of every word.

The title of the review has been updated to the format of the journal

Title

We are grateful for the valuable comments, and we hope we have addressed them adequately for the paper to be considered for publication.

Kind regards,

Martha N Chollom

Reviewer 3 Report

Dear authors,

The paper you present is fascinating and introduces us to the study of water treatment membranes impregnated with silver nanoparticles.

Your work/ review offers very insightful data on the composite membranes impregnated with silver nanoparticles, illustrating the most important aspects regarding their performance and applicability.

It is very clear that you chose an interesting and challenging topic to discuss, which is masterly argued and illustrated through schemes, figures and tables.

I enjoyed reviewing your work and hope it gets published.

Keep us the great work!

Author Response

Manuscript title: "Versatile silver nanoparticles impregnated membranes for water
treatment: a review

Manuscript ID: membranes-2295201

Dear MDPI, Membranes Editorial Office,

We would like to express our gratitude for the comments that reviewers provided concerning our first submission of the manuscript and the opportunity you have given us to revise the manuscript. We have carefully read the comments and revised our manuscript accordingly. The corrections made are highlighted in yellow in the revised manuscript.

Please find a summary of the comments and the actions we took:

Reviewer 3

Reviewer’s comment

Action taken

Section

The paper you present is fascinating and introduces us to the study of water treatment membranes impregnated with silver nanoparticles.

Your work/ review offers very insightful data on the composite membranes impregnated with silver nanoparticles, illustrating the most important aspects regarding their performance and applicability.

It is very clear that you chose an interesting and challenging topic to discuss, which is masterly argued and illustrated through schemes, figures and tables.

I enjoyed reviewing your work and hope it gets published.

Keep us the great work!

We are grateful for the feedback from the reviewer

We are grateful for the valuable comments, and we hope we have addressed them adequately for the paper to be considered for publication.

Kind regards,

Martha N Chollom

Round 2

Reviewer 2 Report

The authors have addressed the suggestions.